# Bee Venom Stimulates Growth Factor Release from Adipose-Derived Stem Cells to Promote Hair Growth

**DOI:** 10.3390/toxins16020084

**Published:** 2024-02-04

**Authors:** Jung Hyun Kim, Tae Yoon Kim, Bonhyuk Goo, Yeoncheol Park

**Affiliations:** 1Department of Acupuncture & Moxibustion, Kyung Hee University Hospital at Gangdong, 892, Dongnam-ro, Gangdong-gu, Seoul 05278, Republic of Korea; 2Department of Traditional Korean Medicine Practice, Jaseng Medical Foundation, 538, Gangnam-daero, Gangnam-gu, Seoul 06110, Republic of Korea; 3Department of Acupuncture & Moxibustion Medicine, Kyung Hee University College of Korean Medicine, Kyung Hee University Hospital at Gangdong, 26, Kyungheedae-ro 4-gil, Dongdaemun-gu, Seoul 02453, Republic of Korea

**Keywords:** bee venom, growth factor, adipose-derived stem cell, hair growth, alopecia

## Abstract

Limited evidence suggests that stimulating adipose-derived stem cells (ASCs) indirectly promotes hair growth. We examined whether bee venom (BV) activated ASCs and whether BV-induced hair growth was facilitated by enhanced growth factor release by ASCs. The induction of the telogen-to-anagen phase was studied in mice. The underlying mechanism was investigated using organ cultures of mouse vibrissa hair follicles. When BV-treated ASCs were injected subcutaneously into mice, the telogen-to-anagen transition was accelerated and, by day 14, the hair weight increased. Quantitative polymerase chain reaction (qPCR) revealed that BV influenced the expression of several molecules, including growth factors, chemokines, channels, transcription factors, and enzymes. Western blot analysis was employed to verify the protein expression levels of extracellular-signal-regulated kinase (ERK) and phospho-ERK. Both the Boyden chamber experiment and scratch assay confirmed the upregulation of cell migration by BV. Additionally, ASCs secreted higher levels of growth factors after exposure to BV. Following BV therapy, the gene expression levels of alkaline phosphatase (ALP), fibroblast growth factor (FGF)-1 and 6, endothelial cell growth factor, and platelet-derived growth factor (PDGF)-C were upregulated. The findings of this study suggest that bee venom can potentially be utilized as an ASC-preconditioning agent for hair regeneration.

## 1. Introduction

The hair follicle regenerative cycle involves growth (anagen), regression (catagen), and resting (telogen) phases before returning to the anagen phase [1]. The dermal papilla (DP) within each follicle plays a crucial role not only in providing nutrients but also in influencing this cyclic process [2]. Adipose-derived stem cells (ASCs) located in protrusions within follicles can differentiate into melanocytes or keratinocyte precursors under appropriate conditions [3].

Hair loss, a condition that arises due to problems in the normal hair production process leading to a decline in hair growth function, is an issue of significant global concern. In 2017 alone, the number of patients seeking treatment for hair loss in countries including the United States, United Kingdom, France, Germany, Italy, Spain, and Japan was estimated at 75 million, with an additional 20 million new diagnoses within the year [4]. Despite existing treatments such as topical minoxidil and oral finasteride medications or even more invasive procedures such as hair transplantation surgeries, approximately 20 million patients remain unresponsive to these traditional methods [5].

While both minoxidil and finasteride have demonstrated effectiveness in treating alopecia specifically linked to androgenic causes [6], they present a considerable risk of recurrence and are less effective for other types of alopecia. These findings highlight the growing need for better therapeutic approaches. One potential approach is the use of bee venom (BV), which has been widely used in traditional Korean medicine owing to its immune-enhancing, circulation-improving, and anti-inflammatory properties [7]. Bee venom is primarily composed of melittin, which accounts for about 50% of its peptides; has a dual nature, being both hydrophilic and hydrophobic; and can damage cell membranes, leading to cell lysis. Other components include apamin, a neurotoxin that makes up less than 2% of the venom and can block calcium-activated potassium channels; the mast cell degranulation peptide, a 2–3% component that promotes inflammation; and adolapin, a minor component that has anti-inflammatory, analgesic, and antipyretic effects [8]. The mechanisms underlying these effects are believed to involve melittin, a component of BV that inhibits immune enzymatic activity by binding to phospholipase A_2_ and works synergistically with apamin to stimulate cortisol secretion [9].

ASCs possess the potential to differentiate into various cell types, including chondrocytes, fibroblasts, adipocytes, and hepatocytes, among others. They release several growth factors, such as vascular endothelial growth factor (VEGF) and β-fibroblast growth factor (FGF), which stimulate dermal papilla cell proliferation, thereby promoting hair growth [10]. ASC transplantation has been shown to promote in vivo hair growth and its conditioned medium increases the proliferation of constituent cells within hair follicles under in vitro conditions [11]. However, it is insufficient to expect that ASC transplantation alone as a single intervention will have a significant effect. Further evidence is needed regarding the validation of potential upregulators on ASCs for hair growth.

Several approaches have been used to discover possible upregulators on ASCs for hair growth. Dermal-papilla-cell-derived extracellular vesicles (DPC-EVs) were found to have varying miRNA expression patterns at different passages [12]. DPC-EVs, in combination with a specialized medium, promoted the proliferation of ASCs and induced the expression of hair-inductive genes such as versican (vcan), alpha-smooth muscle actin (α-sma), osteopontin (opn), and N-Cam (ncam). The expression of hair-inductive genes, proteins, and miRNAs in ASCs treated with DPC-EVs showed similarities to dermal papilla cells (DPCs), indicating that early-passage DPC-EVs can enable ASCs to transdifferentiate into DPC-like cells. DPC-EVs were successfully taken up by ASCs, as confirmed by an EV uptake assay using anti-CD63 antibodies. This study also examined the CTNB signaling pathway in ASCs treated with DPC-EVs to further characterize their properties. Overall, the results suggest that DPC-EVs can act as upregulators of hair-growth-related gene expression in ASCs, potentially enabling them to acquire dermal-papilla-like properties.

Vitamin C has demonstrated the ability to enhance the survival, proliferation, and hair-regenerative potential of ASCs in a dose-dependent manner [13]. This effect is attributed to the upregulation of proliferation-related genes and the facilitation of the telogen-to-anagen transition in mice. Notably, Vitamin C treatment has been found to increase the mRNA expression of growth factors, including HGF, IGFBP6, VEGF, bFGF, and KGF, which are known to play a crucial role in mediating hair growth promotion. Furthermore, Vitamin C has been shown to activate the MAPK pathway and induce ERK1/2 phosphorylation, thereby facilitating ASC proliferation. Inhibition of the MAPK pathway has been observed to attenuate the proliferation of ASCs. Additionally, Vitamin C preconditioning has been found to enhance the hair-growth-promoting effect of ASCs in vivo. Considering its safety, efficacy, affordability, and ease of handling, Vitamin C holds promise as a supplement for ASC cultivation, effectively increasing their yield and regenerative potential.

Despite significant efforts in previous studies, the comprehensive identification of candidate substances that can effectively act on ASCs and positively impact the hair regeneration cycle has proven challenging. The intricate nature of the hair growth cycle, its complex interactions with various substances, and the need for specificity in substance identification contribute to these difficulties. However, the researchers involved in this study remain optimistic about their progress. They believe that BV, which they have been investigating as a potential candidate, may hold the key to bridging this knowledge gap. Preliminary data and observations gathered during the aforementioned studies [12,13] support their anticipation.

While the complete picture and final results are yet to be confirmed, we maintain a hopeful outlook. We remain optimistic that BV has the potential to revolutionize our understanding of the hair regeneration cycle and advance the field significantly. This optimism is not based on mere assumptions but stems from careful scientific analysis and observation. This anticipation paves the way for further research, offering new avenues to explore the role of different substances in the hair regeneration cycle. It represents a significant step forward in the field of hair regeneration research.

This study aims to examine whether BV can indirectly stimulate hair growth through ASCs, despite existing evidence indicating their role in advancing through different phases of the hair follicle cycle. Specifically, we investigate whether BV enhances follicular cell activation through the stimulation of secretion.

## 2. Results

### 2.1. Cell Viability

Following the 3-(4,5-dimethylthiazol-2-yl)-2,5-diphenyl-2H-tetrazolium bromide (MTT) assay results for human ASCs cultured for 24 h with BV at concentrations of 0, 500, 1000, 1500, 2000, and 2500 ng/mL, it was found that BV did not exhibit significant cytotoxicity up to a concentration of 2500 ng/mL (Figure 1). Based on these findings, the concentrations of BV were set at 1 µg/mL and 2 µg/mL for further experiments.

### 2.2. BV-Pretreated ASCs Promote Hair Growth In Vivo

Hair development was upregulated in ASCs [14]. Although ASCs pretreated with BV caused strong hair growth, injecting naïve (untreated) human ASCs minimally promoted telogen-to-anagen induction in mice after subcutaneous injection (Figure 2A,B). We used immunofluorescence labeling for Ki67, a proliferating cell marker in dermal papilla cells, and hematoxylin and eosin (HE) staining to investigate the impact of ASC pretreatment with BV on hair follicles. Compared to mice treated with vehicle or ASC^Ctrl^, the skin of mice treated with ASC^BV^ exhibited a greater number of developed hair follicles (Figure 2C). Furthermore, in contrast to mice treated with vehicle or ASC^Ctrl^, most hair follicles in ASC^BV^-treated mice displayed DP with Ki67+ cells (Figure 2D,E). This finding implies that BV may promote the induction of telogen in the anagen phase.

### 2.3. BV Can Induce Cell Migration in ASCs

To investigate the effect of BV on ASC migration, we conducted both scratch and Boyden chamber assays. BV at 1 µg/mL and 2 µg/mL promoted the migration of ASCs in both the scratch wound and Boyden chamber assays in a dose-dependent manner (Figure 3A,B).

We examined the effects of BV on ASC angiogenesis from ASCs by comparing their in vitro proliferation (Figure 3C). In summary, these results suggest that BV promotes hair growth by upregulating cell migration and ASC-dependent proliferation.

### 2.4. PDGF-C, ECGF, FGF-1, FGF-6, and ALP Induce Hair Growth

Growth factors secreted by ASCs, including platelet-derived growth factor-C (PDGF-C), endothelial cell growth factor (ECGF), fibroblast growth factor (FGF)-1, FGF-6, and alkaline phosphatase (ALP), promote hair follicular stem cell activity and induce the anagen phase in vivo [15,16,17,18]. After qPCR, BV increased the expression of PDGF-C, ECGF, FGF-1, FGF-6, and ALP in comparison to untreated ASCs (Figure 4).

We injected recombinant BV protein into the subcutaneous dermis of shaved mice to examine whether these factors trigger the anagen phase of the hair cycle in vivo. The length of the isolated mouse vibrissal hair follicles in the organ culture was also enhanced by BV treatment (Figure 5A,B). These findings strongly imply that BV stimulates hair growth by releasing growth factors from the ASCs.

### 2.5. BV Can Activate ERK Pathway in ASCs

To investigate the effects and mechanisms of BV on the migration, proliferation, and growth factor secretion of ASCs, we conducted a Western Blot analysis of the ERK pathway and p-ERK pathway. In BV-treated ASCs, both ERK and p-ERK protein levels were upregulated in a dose-dependent manner (Figure 6).

## 3. Discussion

The results of our study are consistent with those of prior research, illustrating the multifaceted impact of BV on a range of cell types, including critical dermal papilla cells and ASCs involved in hair growth and cycling. BV could augment the size of the dermal papilla and stimulate hair growth, potentially via the inhibition of 5α-reductase activity [19]. Although their study focused on the direct application of BV, our findings suggest that BV can also stimulate the release of growth factors from ASCs, thus offering an indirect pathway for enhancing hair growth. Moreover, BV induces apoptosis in melanoma cells via reactive oxygen species [20]. This finding underlines the intricate mechanisms by which BV functions at the cellular level and prompts intriguing questions regarding its potential effects on papilla cells and ASCs under varying conditions or concentrations.

PDGF-C is a member of the PDGF family of proteins. This family of proteins is central to cell growth, division, and development. A study had suggested a possible influence of PDGFs, including PDGF-C, on hair follicle development and cycling. Adipocyte-lineage cells enhance hair growth through PDGF secretion [21]. While this upregulation does not specifically mention the role of PDGF-C in hair growth, given its function within the same family as PDGF-A and its expression in stem cells, which contribute to tissue regeneration similar to that found within skin/hair follicles [22], it could be hypothesized that it may also play a role.

ECGF promotes hair follicle proliferation and differentiation, which are the key processes in hair growth [23]. This aligns with the finding that ECGF enhances the activity of dermal papilla cells, the primary cellular components responsible for regulating the hair growth cycle [24]. Furthermore, these findings are consistent with broader research indicating that factors influencing endothelial cell function, including the integumentary system, can have profound effects on tissue regeneration [25]. Thus, our study highlights the potential therapeutic value of ECGF in conditions associated with impaired hair growth and alopecia. Further investigations are warranted to elucidate the precise molecular mechanisms through which ECGF influences hair follicle biology and to explore its potential applications in dermatological therapeutics.

FGF-1 and FGF-6 have also been identified as influencing factors. Both FGF-1 and FGF-6 have been found to stimulate hair follicle proliferation, contributing to the anagen phase of the hair cycle [26]. The overexpression of FGF-6 in transgenic mice leads to a significant increase in hair follicle density [27]. Moreover, FGF-1 plays a crucial role in promoting angiogenesis around the follicular bulb, thereby enhancing nutrient supply and supporting hair growth [28]. These findings underscore the potential therapeutic value of harnessing these growth factors in conditions related to impaired hair growth or alopecia. Further research is necessary to understand the precise molecular mechanisms and potential applications in dermatological therapeutics.

ALP is an important biomarker. ALP is highly expressed in the dermal papilla cells of hair follicles and its activity correlates with the anagen phase of the hair cycle [29]. ALP activity can be used as an indicator of active hair growth [30]. Furthermore, enhanced ALP activity in papilla cells can promote hair growth, suggesting their potential therapeutic value in treating conditions related to impaired hair growth or alopecia [31]. However, further investigations are needed to elucidate the precise molecular mechanisms by which ALP influences hair follicle biology and to explore its potential applications in dermatological therapeutics.

Our findings highlight the significant role of ASCs and their regulators in modulating extracellular-signal-regulated kinase (ERK) and phosphorylated ERK (p-ERK) pathways, which are crucial for cell survival, proliferation, and differentiation. Previous studies have shown that ASCs can enhance tissue regeneration by activating various signaling pathways, including ERK/p-ERK [32]. Moreover, it has been demonstrated that the upregulation of growth factors can augment ASC activity by activating the ERK/p-ERK pathway, leading to enhanced cell proliferation and differentiation [33]. The present study supports these findings by demonstrating a significant increase in ERK and p-ERK levels after treatment with ASCs and their upregulators. This suggests a potential therapeutic approach for enhancing tissue regeneration through targeted modulation of this pathway.

## 4. Conclusions

In conclusion, the telogen-to-anagen transition in mice was induced by subcutaneous injection of BV-treated ASCs, and BV treatment enhanced ASC migration, tube formation, and growth factor secretion. Each of the most highly elevated growth factors, PDGF-C, ECGF, FGF-1, FGF-6, and ALP, enhanced anagen induction in mice. Thus, BV exhibits substantial potential applications as a novel ASC-preconditioning agent for augmenting hair growth.

## 5. Materials and Methods

### 5.1. Adiopose-Derived Stem Cells

The human ASCs used in this research were derived from human tissue and purchased from ATCC (Manassas, VA, USA). The cells were produced on 28 October 2020. The cells were donated by a 49-year-old African American female. Informed consent was obtained by ATCC itself. The donor tissue source was adipose, or fat tissue. The cells were cryopreserved in the appropriate medium with a fill volume of 1.1 mL.

### 5.2. Cell Culture

Initially, ASCs were cultured in KSB-3 Basal Medium (Kang Stem BIOTECH, Seoul, Republic of Korea) supplemented with 10% fetal bovine serum (FBS: Gibco, Waltham, MA, USA) and 1% antibiotic-antimycotic (Gibco). This culturing process was carried out in a controlled environment set at a temperature of 37 °C and an atmosphere of 5% CO₂. In the subsequent step, to induce the growth of HASC, the cells were cultured for 24 h in a medium supplemented with BV. BV powder was purchased from Chung Jin Biotech Co., Ltd. (Ansan, Republic of Korea). This methodology facilitated adequate growth and differentiation of cells.

### 5.3. Cytotoxicity Assessment

Human ASCs were cultured at a density of 1 × 10^5^ cells/well in a 96-well plate. The samples were cultured at various concentrations. After 24 h of culture, 100 μL of (3-(4,5-dimethylthiazol-2-yl)-2,5-diphenyltetrazolium bromide) MTT solution was added to each well and incubated for an additional hour at 37 °C. Absorbance was then measured using an ELISA reader at a wavelength of 540 nm.

### 5.4. Scratch Migration Assay

ASC cells were seeded in a six-well plate at a density of 1 × 10^5^ cells/well. When cell confluence reached approximately 80–90%, the medium was replaced with a serum-free medium to induce starvation for a period of four hours. A scratch was then made in each well using a 1000 µL pipette tip, followed by treatment with BV, diluted at various concentrations, and cultured for an additional 96 h. The progress of the cells was documented at the 48, 72, and 96 h marks using a digital color microscopy camera (Carl Zeiss, Jena, Germany). Images were analyzed using the ImageJ software (https://imagej.net/ij/ accessed on 1 November 2023) (National Institutes of Health, Bethesda, MD, USA) to quantify cell closure.

### 5.5. Boyden Chamber Assay

The migratory and invasive capabilities of each cell line were compared using the Boyden chamber assay. A 60 μg/mL concentration of matrigel (Collaborative Research, Bedford, MA, USA) was applied dropwise to a polycarbonate membrane filter and allowed to dry overnight at 37 °C. The lower chamber was filled with cell culture medium, and the filter was placed and securely capped. Cells were harvested by trypsin-EDTA (GIBCO) treatment and resuspended in a single-cell suspension for counting. The upper chamber was filled with 5 × 10^4^ cells. After incubation at 37 °C for 12–16 h, the filter was fixed in formaldehyde and stained with Giemsa for 15 min. After rinsing under running water, the underside of the filter was swabbed thrice with a cotton swab to remove non-migratory cells. The filters were placed in a 24-well plate and dried before counting the number of stained cells that adhered to the underside of each filter.

### 5.6. Cell Growth Assay

To measure the effect of BV on ASC growth, the cells were seeded in a 12-well plate at a density of 1 × 10^5^ cells/well. BV was then diluted to various concentrations and added to the wells, after which the cells were cultured for a period of seven days. Cell viability was assessed by trypan blue staining and living cell counts were subsequently calculated.

### 5.7. Animal Experiment

Twenty male C57BL/6 mice, six weeks old, and approximately 20 g in weight, were procured from Raon Bio (GemPharmatech, Gyeonggi-do, Republic of Korea). These animals were provided with free access to commercial chow (DooYeol Biotech, Seoul, Republic of Korea) and tap water. To acclimatize them to the experimental conditions, they were housed for one week in a thermo-hygrostat maintained at a temperature of 21 ± 2 °C and humidity of 55 ± 3%.

The mice were divided into three experimental groups: a control group (five mice), an ASC cell injection group (five mice each receiving an injection of 3 × 10^4^ cells), and a third group that received both ASC cells (3 × 10^4^) and BV (five mice). Before the start of the experiment, the hair at the intended site of the intervention was removed using an animal-specific clipper. Any remaining hair was removed using a depilatory cream (Nikrin, IlDong Pharmaceuticals, Seoul, Republic of Korea). A recovery period of 24 h was allowed before the start of the experiment.

The two-week-long experiment involved different injections for each group: normal saline for the control group; ASC cells (3 × 10^4^) for the cell injection group; and a mixture of ASC cells (3 × 10^4^) and BV (ChungJin Biotech, Ansan, Republic of Korea) at a concentration of 2 µg/mL for the third group. Progression was documented photographically on days 0, 7, 10, and 14.

### 5.8. Hematoxylin/Eosin and Immunofluorescence Staining

To observe hair growth, the dorsal skin of mice was excised using sterile surgical scissors. The excised skin tissues were fixed in 10% formalin for 24 h and washed under running water. Skin tissues were cut at intervals of 4 mm and placed in tissue capsules. They were then dehydrated by treating with a dilution series of alcohol, including 50, 70, 80, 95, and 100% alcohol for one hour each. The samples were then immersed in xylene four times, each time for an hour, followed by paraffin substitution.

The skin tissues were embedded in paraffin and sections with a thickness of 4 μm were prepared from the embedded tissues. The sections were stained with hematoxylin and eosin (Sigma-Aldrich, St. Louis, MO, USA) and observed under a light microscope at ×40.

### 5.9. Ki+ and DAPI Staining

For immunofluorescence staining, the cells were fixed with 4% paraformaldehyde for 15 min at room temperature. Following fixation, the cells were permeabilized with 0.1% Triton X-100 in PBS for 10 min and blocked with a solution of 5% bovine serum albumin (BSA) in PBS for one hour to prevent nonspecific binding. The cells were then incubated overnight at 4 °C with the primary antibody against Ki67 (1:200 dilution; Abcam, Waltham, MA, USA), a marker of cell proliferation.

After washing three times with PBS, the cells were incubated with Alexa Fluor^®^ 488-conjugated secondary antibody (1:500 dilution; Invitrogen, Seoul, Republic of Korea) for one hour at room temperature in the dark. Subsequently, to visualize the nuclei, the samples were stained with DAPI (4′,6-diamidino-2-phenylindole; Sigma-Aldrich) at a concentration of 300 nM for 5 min. Next, the samples were washed three times with PBS and mounted on glass slides using an anti-fade mounting medium.

Images of the stained samples were captured using a fluorescence microscope equipped with the appropriate filters. The number of Ki67-positive cells was counted manually in five random fields per sample and normalized to the total number of DAPI-stained nuclei to obtain the percentage of proliferating cells.

### 5.10. RNA Extraction and Quantitative RT-PCR

We hypothesized that BV could indirectly promote hair growth by enhancing the release of these growth factors from ASCs. To explore this possibility, we compared the expression patterns of untreated naïve ASCs and BV-treated ASCs in vivo, using quantitative polymerase chain reaction (qPCR). Total RNA was extracted from the samples using TRIzol reagent (Invitrogen, Thermo Fisher Scientific, Seoul, Republic of Korea), and cDNA was synthesized according to the manufacturer’s protocol using the RevertAid First Strand cDNA Synthesis Kit (Thermo Fisher Scientific). The reaction mixture for PCR consisted of 1 μL of synthesized cDNA, 10 μL of Power SYBR Green PCR master mix (2×) (Applied Biosystems, Waltham, MA, USA), primers, and DEPC-treated distilled water. The PCR conditions were set at 95 °C for 15 s, followed by 60 °C for 30 s, and then 72 °C for 30 s repeated for 40 cycles. The primer sequences used in the experiments are shown in the Appendix A.

### 5.11. Western Blot

Samples were lysed in 50 μL of 1× RIPA buffer, and protein concentration was determined using the Bio-Rad DC Protein Assay. Samples containing equal amounts of protein were prepared using 5× lane marker-reducing sample buffer. The prepared samples were separated using SDS-PAGE using a 10% polyacrylamide gel, and then transferred to a nitrocellulose membrane at 230 mA for 90 min at 4 °C. Primary antibodies including rabbit anti-phospho-p44/42 (1:1000; Cell Signaling Technology, Danvers, MA, USA), rabbit anti-p44/42 (1:1000; Cell Signaling Technology), and rabbit anti-actin (1:1000; Santa Cruz Biotechnology, Santa Cruz, CA, USA) were incubated with the membrane overnight at 4 °C. The secondary antibody was horseradish-peroxidase-conjugated anti-rabbit IgG (1:5000; Cell Signaling Technology), which was incubated for 1 h at around 28 °C. Proteins were visualized using Pierce ECL Western Blotting Substrate (Thermo Fisher Scientific) or SuperSignal West FemtoMaximum Sensitivity Substrate (Thermo Fisher Scientific). The membranes were stripped with Restore Western Blot Stripping Buffer for probing with different antibodies.

### 5.12. Statistical Analysis

Each experiment was performed in separate cultures more than three times. The data are shown as mean plus standard error (SEM). Student’s *t*-test was used to compare the means. A 0.05 level of confidence was considered statistically significant for all statistical tests. However, Bonferroni adjustments to alpha levels were applied if needed. In these cases, when two tests were run, 0.025 became the critical value instead of 0.05. If the more proper outcome could no longer be interpreted as statistically significant after Bonferroni correction, the exact *p*-value was provided.

### 5.13. AI Language Model Usage

In the course of crafting the manuscript, the use of multiple artificial intelligence (AI) resources played a crucial role in confirming the accuracy and clarity of the English language utilized. Grammarly (https://www.grammarly.com/, accessed on 1 November 2023), an advanced tool powered by AI, was brought into play for thorough inspections of grammar, spelling, and punctuation. The Hemingway Editor (http://www.hemingwayapp.com/, accessed on 1 November 2023) was applied to increase the understandability of the text and to lessen the intricacy of sentences. LanguageTool (https://languagetool.org/, accessed on 1 November 2023), adept at performing grammar and spelling checks that are context-specific across a diversity of English dialects, was implemented as well. Ginger (https://www.gingersoftware.com/, accessed on 1 November 2023), another tool with AI, was employed for the restructuring of sentences and the enhancement of language flow. Slick Write (https://www.slickwrite.com/, accessed on 1 November 2023) was brought in to carry out an analytical examination of sentence construction and the overarching writing style. These tools were used in a repetitive manner throughout the writing process to uphold a high caliber of academic English.

## Figures and Tables

**Figure 1 toxins-16-00084-f001:**
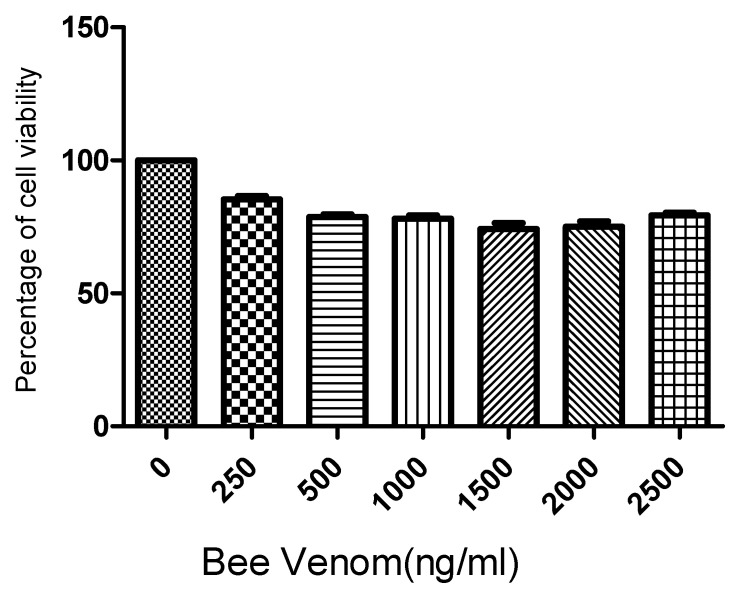
Results (mean + SEM) of MTT (tetrazolium dye) assay to assess toxicity of bee venom to adipose stem cells (ASCs). N = 1 × 10^5^ cells/well for each mean. MTT: 3-(4,5-dimethylthiazol-2-yl)-2,5-diphenyl-2H-tetrazolium bromide.

**Figure 2 toxins-16-00084-f002:**
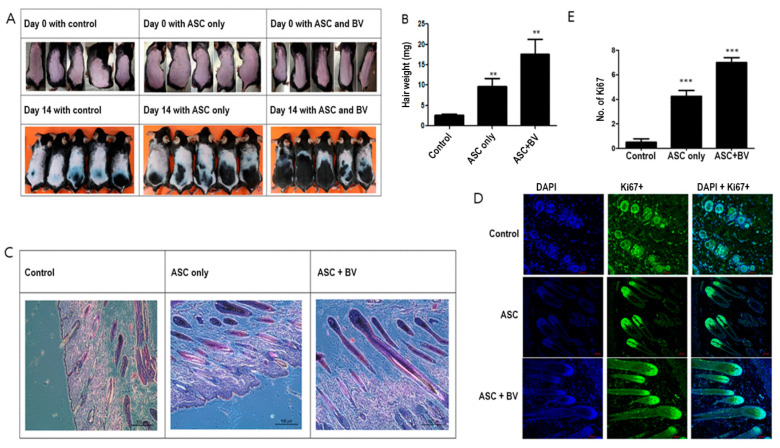
(**A**) Changes in mouse hair at 14 days with adipose-derived stem cells (ASCs) and bee venom (BV). (**B**) Hair weight changes (mean + SEM) after ASC and ASC + BV pretreatment, N = 32 for each mean, ** *p* < 0.01 for ASC only relative to control and ASC + BV relative to ASC only. (**C**) The effect of ASC and ASC + BV pretreatment demonstrated by HE staining. (**D**) The effect of ASC and ASC + BV pretreatment demonstrated by DAPI and Ki67+ staining. (**E**) The number of Ki67+ cells (mean + SEM) after pretreatment with ASC and ASC + BV, N = 32 for each mean, *** *p* < 0.001 for ASC only relative to control and ASC + BV relative to ASC only.

**Figure 3 toxins-16-00084-f003:**
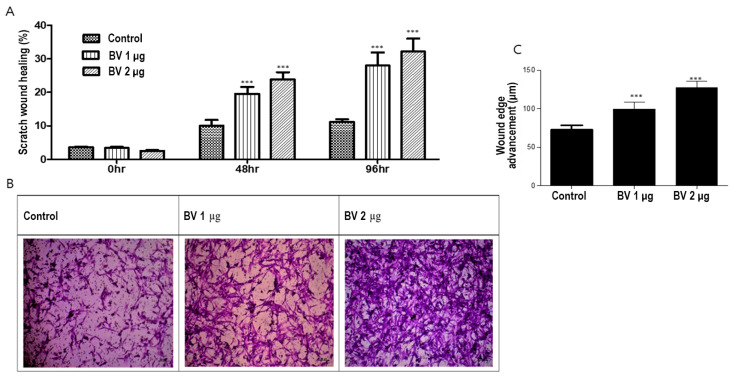
(**A**) Results (mean + SEM) of scratch assay with bee venom (BV), N = 31 for each mean, *** *p* < 0.001 for BV 1 µg relative to control and BV 2 µg relative to BV 1 µg. (**B**) Results of Boyden chamber assay with BV. (**C**) The effect (mean + SEM) of BV on in vitro angiogenesis proliferation, N = 31 for each mean, *** *p* < 0.001 for BV 1 µg relative to control and BV 2 µg relative to BV 1 µg.

**Figure 4 toxins-16-00084-f004:**
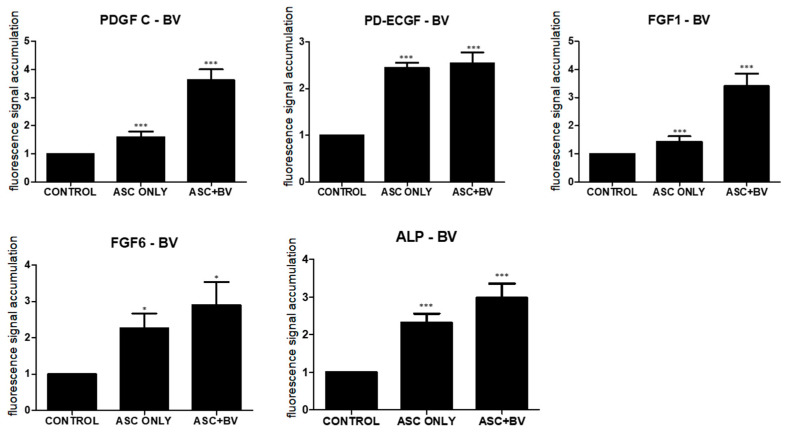
Results (mean + SEM) of qPCR in vivo, N = 31 for each mean; * *p* = 0.027, approaching significance after Bonferroni correction; *** *p* < 0.001; ASC only is compared to control and ASC + BV is compared to ASC only. The adipose stem cell + bee venom (ASC + BV) group is statistically relevant to the ASC-only group.

**Figure 5 toxins-16-00084-f005:**
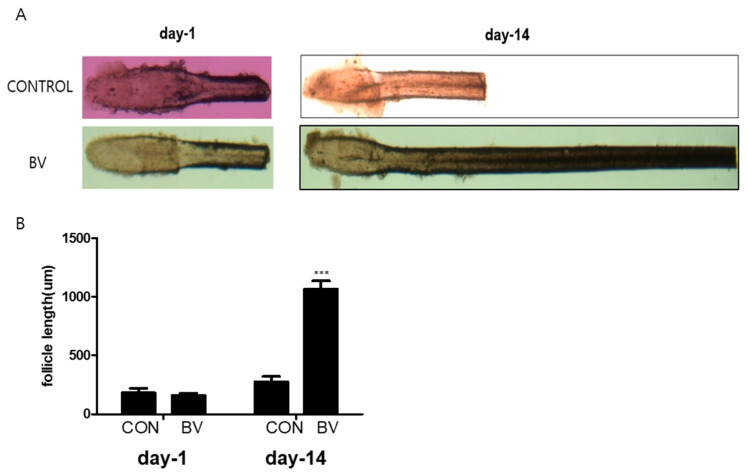
(**A**) Vibrissal hair follicle morphological changes after 1 and 14 days of bee venom (BV) treatment. (**B**) Follicle length changes (mean + SEM) after 1 and 14 days of BV treatment, N = 29 for each mean, *** *p* < 0.001 relative to control (CON).

**Figure 6 toxins-16-00084-f006:**
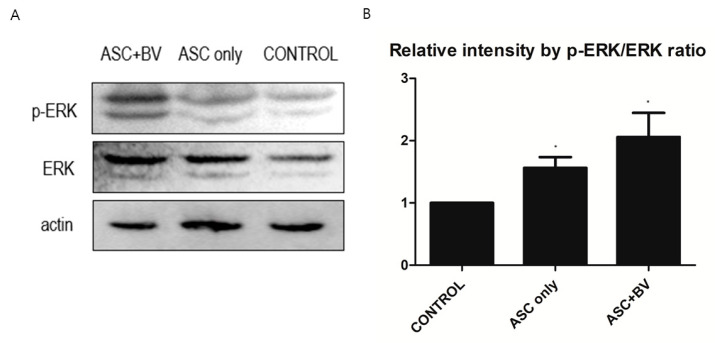
(**A**) Western blot results of apical stem cell (ASC) and ASC + bee venom (ASC + BV), N = 32 for each mean. (**B**) Protein expression levels (mean + SEM) demonstrated with extracellular-signal-regulated kinase (ERK) and phosphorylated ERK (p-ERK) pathways, N = 29 for each mean, * *p* = 0.031, which approached significance after Bonferroni correction.

## Data Availability

Data can be obtain upon request from the corresponding author.

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
