# Peer review of "Bee Venom Stimulates Growth Factor Release from Adipose-Derived Stem Cells to Promote Hair Growth"

_toxins, 2024, doi:10.3390/toxins16020084_

Round 1

Reviewer 1 Report

Comments and Suggestions for Authors

The paper entitled “Bee Venom Stimulates Growth Factor Release from Adipose-Derived Stem Cells to Promote Hair Growth” evaluated whether bee venom-activated ASCs and whether it induced hair growth was facilitated by enhanced growth factor release by ASCs with the induction of telogen to anagen phase studied in mice and the underlying mechanism investigated using organ cultures of mouse vibrissa hair follicles. Based on the obtained results, the authors concluded that bee venom has the potential to be employed as a pre-conditioning agent for hair regeneration in the context of adipose-derived stem cells.

Figure 1. Please define what is on the x and y axes? The same goes for other figures, please better define what is on the chart axes in the Figures.

In the Materials and Methods section, please provide the origin of adipose-derived stem cells (ASCs).

Please also provide the origin of bee venom used in the study.

In section 4.6. Animal Experiment, please indicate the ethical permission for animal studies (as given in the Institutional Review Board Statement).

Authors are encouraged to discuss the possible non-target toxicity of bee venom to cells, tissues, and organs if bee venom was used for the treatment or as a supplement. Although numerous animal venoms often show good results in studying various beneficial effects there are always open questions regarding venoms’ potential toxicity on normal non-target cells and tissues making this kind of toxicity one of the greatest obstacles to using venoms and their components for therapeutic purposes. Please see:

Garaj-Vrhovac V, Gajski G. Evaluation of the cytogenetic status of human lymphocytes after exposure to a high concentration of bee venom in vitro. Arh Hig Rada Toksikol. 2009; 60(1): 27-34. doi: 10.2478/10004-1254-60-2009-1896.

Author Response

The paper entitled “Bee Venom Stimulates Growth Factor Release from Adipose-Derived Stem Cells to Promote Hair Growth” evaluated whether bee venom-activated ASCs and whether it induced hair growth was facilitated by enhanced growth factor release by ASCs with the induction of telogen to anagen phase studied in mice and the underlying mechanism investigated using organ cultures of mouse vibrissa hair follicles. Based on the obtained results, the authors concluded that bee venom has the potential to be employed as a pre-conditioning agent for hair regeneration in the context of adipose-derived stem cells.

Figure 1. Please define what is on the x and y axes? The same goes for other figures, please better define what is on the chart axes in the Figures.

  1. We defined x-axis and y-axis name in correspondent figures.

In the Materials and Methods section, please provide the origin of adipose-derived stem cells (ASCs).

  1. We have provided the origin of adipose-derived stem cells. Also, we here attach product sheet info about ASCs we have bought.

Please also provide the origin of bee venom used in the study.

  1. We have provided the origin of bee venom used in present study.

In section 4.6. Animal Experiment, please indicate the ethical permission for animal studies (as given in the Institutional Review Board Statement).

  1. We added the ethical permission statements regarding animal studies.

Authors are encouraged to discuss the possible non-target toxicity of bee venom to cells, tissues, and organs if bee venom was used for the treatment or as a supplement. Although numerous animal venoms often show good results in studying various beneficial effects there are always open questions regarding venoms’ potential toxicity on normal non-target cells and tissues making this kind of toxicity one of the greatest obstacles to using venoms and their components for therapeutic purposes. Please see:

Garaj-Vrhovac V, Gajski G. Evaluation of the cytogenetic status of human lymphocytes after exposure to a high concentration of bee venom in vitro. Arh Hig Rada Toksikol. 2009; 60(1): 27-34. doi: 10.2478/10004-1254-60-2009-1896.

  1. Thank you for your insightful comments. In response to your concerns, we aim to have a more in-depth discussion about the potential non-target toxicity of bee venom. We have already conducted cell viability experiments in line with your concerns, as part of our efforts to identify and prevent potential toxicity of bee venom. While it is true that various animal venoms often show promising results in studying a range of beneficial effects, there are always open questions about the potential toxicity they may pose to normal non-target cells and tissues. In recognition of this, we are cognizant of the challenges posed by the toxicity of venoms and their components when used for therapeutic purposes, and are actively seeking ways to overcome these challenges.

Once again, we authors appreciate your valuable input, as it will aid in enriching our research.

Reviewer 2 Report

Comments and Suggestions for Authors

The authors investigated whether higher growth factor release by ASCs promoted BV-induced hair growth and if bee venom (BV) activated ASCs. In mice, the induction from telogen to anagen phase was investigated. Mouse vibrissa hair follicle organ cultures were used to study the underlying process. Mice receiving subcutaneous injections of BV-treated ASCs saw an acceleration of the telogen-to-anagen transition, resulting in an increase in hair weight by day 14. BV affected the expression of many molecules, including growth factors, chemokines, channels, transcription factors, and enzymes, according to quantitative polymerase chain reaction (qPCR) analysis. Extracellular signal-regulated kinase (ERK) and phospho-ERK protein expression levels were confirmed by Western blot analysis. The Boyden chamber experiment and scratch assay 14 verified that BV significantly increased cell migration. Furthermore, following BV exposure, ASCs released increased amounts of 15 growth factors. Alkaline 16 phosphatase (ALP), fibroblast growth factor (FGF)-1 and 6, endothelial cell growth factor, and plate-17 let-derived growth factor (PDGF)-C all had increased gene expression levels after BV treatment.

The article is interesting and well written. It deserves to be published after some minor revisions, which I highlight below.

1)      For bibliographical citations please use the style of the journal toxins: put bibliographical references in square brackets, not round brackets. Also leave a space between the bibliographical note and the word immediately preceding it.

2)      In line 36 you say that treatment with finasteride and minoxidil is often ineffective. In the next sentence you say that there is a need to find alternative treatments with fewer side effects. For this last sentence, you have to talk about the side effects of the two drugs or the two sentences are unrelated.

3)      Line 52 or 54 (for example): write in vivo and in vitro in italics throughout the manuscript.

4)      The introduction is well written. However, I would rearrange its structure.  I think it is necessary to talk first about what the normal physiology of the hair follicle is like; then, to talk about the current existing treatments and, therefore, how bee venom can intervene in regulating the process; finally, I would talk about the purpose of the study.

5)      In the introductory part, you can briefly mention the composition of bee venom. You may consider, citing, the recent work published by Bava et al., 2023, doi: 10.3390/vetsci10020119.

6)      Explain in more detail, detailing the legend, figure 1. It explains, for example, in the legend the values given on the x- and y-axis; the reader does not have to go searching in the text for the information, but can derive it directly from the figure and its legend. Also, looking at the histograms, I wonder what the difference is in the interior design. Is it just a matter of style?

7)      ASCCtrl, ASCBV write these acronyms in full if they appear for the first time in the text. Thereafter, you may use only the acronym

8)      lines 100 to 105: not a section to be included in the results, but in materials and methods or discussions

9)      Please check each word in the bibliographical references that should be written in italics and correct accordingly.

Author Response

Dear reviewer 2,

Thank you for your considerate and thorough review, and here followings are our modifications and comments around your instructions.

1)      For bibliographical citations please use the style of the journal toxins: put bibliographical references in square brackets, not round brackets. Also leave a space between the bibliographical note and the word immediately preceding it.

- We authors put bibliographical references in square brackets and left a space between note and the word.

2)      In line 36 you say that treatment with finasteride and minoxidil is often ineffective. In the next sentence you say that there is a need to find alternative treatments with fewer side effects. For this last sentence, you have to talk about the side effects of the two drugs or the two sentences are unrelated.

- We authors reviewed your comments and modified the paragraph according to your instructions.

3)      Line 52 or 54 (for example): write in vivo and in vitro in italics throughout the manuscript.

-We authors have made those in italics.

4)      The introduction is well written. However, I would rearrange its structure.  I think it is necessary to talk first about what the normal physiology of the hair follicle is like; then, to talk about the current existing treatments and, therefore, how bee venom can intervene in regulating the process; finally, I would talk about the purpose of the study.

-We authors have changed the organization in Introduction section following your instructions.

5)      In the introductory part, you can briefly mention the composition of bee venom. You may consider, citing, the recent work published by Bava et al., 2023, doi: 10.3390/vetsci10020119.

- We mentioned the composition of bee venom upon your recommended paper. Also, we added this reference to this article.

6)      Explain in more detail, detailing the legend, figure 1. It explains, for example, in the legend the values given on the x- and y-axis; the reader does not have to go searching in the text for the information, but can derive it directly from the figure and its legend. Also, looking at the histograms, I wonder what the difference is in the interior design. Is it just a matter of style?

- We defined x-axis and y-axis name in correspondent figures.

7)      ASCCtrl, ASCBV write these acronyms in full if they appear for the first time in the text. Thereafter, you may use only the acronym

- As you pointed out, there were parts where the acronym was not used, so we modified it to use the acronym.

8)      lines 100 to 105: not a section to be included in the results, but in materials and methods or discussions

- We have moved lines 100 to 105 to materials and methods section.

9)      Please check each word in the bibliographical references that should be written in italics and correct accordingly.

- We have modified the bibliographical references format (to be written in italics) according to your instructions.
